# Genetic Engineering of *Klebsiella pneumoniae* ATCC 25955 for Bioconjugate Vaccine Applications

**DOI:** 10.3390/microorganisms11051321

**Published:** 2023-05-17

**Authors:** Yan Liu, Shulei Li, Yan Guo, Xin Li, Li Zhu, Hengliang Wang, Jun Wu, Chao Pan

**Affiliations:** 1State Key Laboratory of Pathogen and Biosecurity, Beijing Institute of Biotechnology, Beijing 100071, Chinawanghl@bmi.ac.cn (H.W.); 2State Key Laboratory of Biochemical Engineering, Institute of Process Engineering, Chinese Academy of Sciences, Beijing 100190, China

**Keywords:** *Klebsiella pneumoniae*, glycoengineering, PGCT, CRISPR, nanovaccine

## Abstract

Vaccination is considered the most effective means to fight against the multidrug-resistant strains of *Klebsiella pneumoniae*. In recent years, a potential protein glycan coupling technology has been extensively used in the production of bioconjugated vaccines. Here, a series of glycoengineering strains derived from *K. pneumoniae* ATCC 25955 were designed for protein glycan coupling technology. The capsule polysaccharide biosynthesis gene cluster and the O-antigen ligase gene *waaL* were deleted via the CRISPR/Cas9 system to further weaken the virulence of host stains and block the unwanted endogenous glycan synthesis. Particularly, the SpyCatcher protein in the efficient protein covalent ligation system (SpyTag/SpyCatcher) was selected as the carrier protein to load the bacterial antigenic polysaccharides (O1 serotype), which could covalently bind to SpyTag-functionalized nanoparticles AP205 to form nanovaccines. Furthermore, two genes (*wbbY* and *wbbZ*) located in the O-antigen biosynthesis gene cluster were knocked out to change the O1 serotype of the engineered strain into the O2 serotype. Both KPO1-SC and KPO2-SC glycoproteins were successfully obtained as expected using our glycoengineering strains. Our work provides new insights into the design of nontraditional bacterial chassis for bioconjugate nanovaccines against infectious diseases.

## 1. Introduction

*Klebsiella pneumoniae*, as the second leading conditional pathogenic bacterium after *Escherichia coli* in terms of infection rate, has always been a target for vaccine research [1]. In particular, carbapenem-resistant *K. pneumoniae*, which has emerged in the past two decades, further reduces the effectiveness of clinical antimicrobial therapy because of its resistance to penicillins, cephalosporins, carbapenems, monobactams, and penems. Carbapenem-resistant *K. pneumoniae* has rapidly spread and caused outbreaks worldwide, leading to a significant increase in mortality when compared with carbapenem-sensitive strains [2,3]. To address the threat of antimicrobial resistance, vaccination has emerged as the most effective strategy for preventing infection. During the last decades, various candidate vaccines against *K. pneumoniae* have been investigated, including inactivated vaccines, attenuated live vaccines, and subunit vaccines [4]. Among them, the single, quadrivalent, and 24-valent polysaccharide vaccines developed by the WRAIR team have entered clinical trials [5].

The polysaccharide vaccine has a clear drawback of exhibiting poor immunogenicity. Especially within the newborn body, it cannot induce long-term immune memory [6,7]. When polysaccharides are combined with protein carriers, glycoproteins will be transformed into T-cell-dependent antigens, exhibiting stronger immunogenicity [8,9]. Therefore, glycoconjugate vaccines are currently one of the safest and most effective vaccine forms used. The Jain team analyzed the antibody type induced by purified K11 capsule polysaccharides combined with bovine serum albumin. It was observed that K11 capsule polysaccharides alone can only induce a serum IgM response in mice, whereas K11 glycoprotein elicited a serum IgG response [10]. The Seeberger, P.H. team combined bacterial capsule polysaccharide (CPS) with CRM197 protein and demonstrated that the target glycoprotein has good immunogenicity [11]. The Ravinder team produced glycoconjugate vaccines on the basis of the K2 capsule and demonstrated that the antibodies induced by the vaccine exhibited excellent bactericidal effects [12]. Compared with the CPS, the O-antigen polysaccharide (OPS) has fewer serovars, and scientists have demonstrated that antibodies against the OPS are protective against diseases [13,14,15].

In recent years, the functional analysis of bacterial protein glycosylation systems, a synthetic biological method for preparing conjugate vaccines called protein glycan coupling technology (PGCT), have developed rapidly and gradually gained widespread attention [16]. Compared with the classic chemical method, the main advantage of PGCT is that the in vivo biosynthesis process produces a relatively uniform final product, which can be easily obtained after one-step fermentation. This also results in fewer downstream purifications and a more convenient quality control process. PGCT normally requires three key elements: the gene cluster for expression of the antigenic polysaccharide, the oligosaccharyltransferase, and a carrier protein [17,18]. The currently used oligosaccharyltransferases are mainly divided into two categories: *N*-glycosyltransferases and *O*-glycosyltransferases. PglB is representative of *N*-glycosyltransferases and has been extensively used in the field of *K. pneumoniae* vaccines [19,20]. For example, the Feldman team used it to connect K1 and K2 CPS serotypes of *K. pneumoniae* with carrier protein EpA in recombinant conjugate vaccines prepared using bioengineered *E. coli* cells. The results indicated that after undergoing an immunization, mice can survive infections from the two main serotypes of hypervirulent *K. pneumoniae* [21].

Although model microorganisms such as *E. coli* have been extensively developed and used as chassis strains for the production of various compounds, certain microorganisms possess natural advantages over *E. coli* in terms of growth rate and protein expression, or they possess some metabolic pathways can lead to better efficiency for specific purposes. *K. pneumoniae* strain ATCC 25955, as a low-virulence O1 serotype strain, can naturally synthesize various compounds and has been engineered by some teams for the production of compounds such as 1,3-propanediol. It can metabolize cost-efficient crude glycerol, which greatly reduces the fermentation costs. Moreover, it has been reported that the growth rate of this strain is more than 30% faster than that of *E. coli* under the same conditions, rendering it a suitable strain for glycoengineering [22,23,24].

Therefore, several glycoengineering bacterial strains derived from *K. pneumoniae* strain 25955 were constructed using the CRISPR/Cas9 tools and were screened to identify the most suitable one to produce the target glycoprotein-bearing O1 antigenic polysaccharide in this study. Notably, the SpyCatcher protein (SC) in the bioglue SpyTag/SpyCatcher system was selected as the carrier protein, which could spontaneously form an isopeptide bond with SpyTag-functionalized nanoparticles to form a nanovaccine. A nanovaccine generally refers to a new generation of vaccines for disease prevention or treatment using nanoparticles (NPs) as transport carriers and/or adjuvants with a size of approximately 20–250 nm [25]. Nanovaccines have exhibited great potential because of their many advantages such as high antigen-loading capacity, targeted delivery, and effective stimulation of the immune system. Considering that O1 and O2 serotypes account for more than 60% of clinical isolates of carbapenem-resistant and broad-spectrum β-lactam-resistant bacteria [24], we also further transformed the serotypes of the 25955 glycoengineered chassis into O2 serotypes. This work provides at least two safe and effective engineered *K. pneumoniae* strains for the production of bioconjugate nanovaccines. The strategy described in this study is also applicable to vaccines against other infectious diseases.

## 2. Materials and Methods

### 2.1. Strains and Plasmids

The strains and plasmids used in this study are listed in Table 1. All strains in this study were sequentially mutated and genomically modified from the *K. pneumoniae* strain 25955 using CRISPR gene editing technology. All primers are presented in Appendix A. All bacterial strains were cultured in Luria–Bertani (LB; peptone 10 g/L, yeast extract 5 g/L, and NaCl 10 g/L) broth or on solid LB medium containing 1.5% agar (antibiotics added as necessary to maintain the plasmid long term: kanamycin 50 μg/mL, ampicillin 50 μg/mL, chloramphenicol 50 μg/mL, and spectinomycin 50 μg/mL).

### 2.2. CRISPR

The gene mutations were performed through the CRISPR method, which has been reported by Jiang et al. [28]. Briefly, the N20 portion of the single guide RNA was designed using the online N20 design tool (http://crispor.tefor.net/, accessed on 8 May 2023) [29], and then the designed N20 was constructed into the constitutively expressed plasmid pTargetF (Addgene 62226) using the NEBridge^®^ Golden Gate Assembly Kit (NEB #E1602). The sequence was verified to be correct by sequencing. To obtain the recombinant template, the 750 bp upstream and downstream sequences of the deletion cluster were selected and linked together using the NEBridge^®^ Golden Gate Assembly Kit (NEB #E1602). The target strain was prepared by electroporation of 400 ng of pCas plasmid (Addgene 62225). One hour before the preparation of the competent cells, 2% arabinose was added to the strain to induce the λ Red recombination system, which was constructed in the plasmid pCas. Then, 300 ng of the constructed pTargetF plasmid and 800 ng of the recombinant template DNA fragment were electroporated into the strain. After the cultures were grown overnight on an LB plate, the correct editing of the strain was verified by polymerase chain reaction (PCR) and sequencing (Appendix A). Subsequently, edited colonies were cultured overnight at 30 °C in LB with kanamycin and 1 mM of Isopropyl β-D-Thiogalactoside (IPTG), serially diluted, and plated on kanamycin plates. The spot plates with kanamycin and spectinomycin were incubated overnight at 30 °C to identify colonies that have lost pTarget (no spc resistance) but retained pCas (Kan resistance). At this stage, the strains were either induced with 2% arabinose and again made capable of further mutation, or the pCas was removed from the strains by growing overnight at 42 °C without antibiotics.

### 2.3. Glycoprotein-Induced Expression

To express the glycoprotein, glycerol bacteria cryopreserved in a −80 °C refrigerator were used. In a biosafety cabinet, a tip was used to transfer and streak bacteria onto an LB plate. The overnight culture was performed in a 37 °C incubator. Monoclonal clones were picked the following day, and PCR was used to verify the loss of the expression plasmid. The verified correct clone was inoculated into 5 mL liquid LB and incubated overnight at 37 °C and 220 rpm. The next day, the bacterial suspension was inoculated into LB liquid medium at a ratio of 1:100 and incubated at 37 °C and 220 rpm until OD_600_ = 0.6–0.8. Then, 1 mol/L of IPTG (for a final concentration of 1 mM) was added and incubated overnight at 30 °C and 220 rpm on a shaker. The samples were collected and processed, and their expression was verified.

### 2.4. Western Blot

Western blot was performed as described previously [27]. Horseradish peroxidase–conjugated 6× His Tag antibody (Abmart Shanghai Co., Ltd., Shanghai, China) was used to detect proteins with a 6× His tag. If there were more nonspecific bands, the 6× His antibody was incubated with *E. coli* W3110 cell lysate for 1 h before detection. The antibodies against *K. pneumoniae* serotypes O1 and O2 were obtained by immunizing Japanese white rabbits with *K. pneumoniae* strain 355 and *K. pneumoniae* strain 041 whole bacteria, respectively, which were used to detect the glycan fraction of glycoproteins. Horseradish peroxidase-conjugated goat anti-rabbit IgG (Transgen Biotech, Inc., Beijing, China) was used as the secondary antibody.

### 2.5. Protein Purification

The induced cells were collected by centrifugation at 7104 *g* for 10 min and then resuspended with buffer A (30 g NaCl, 13.3 mL Tris-HCl (pH 8.8), 0.6 g imidazole, and 1 mL Tween-20). Then, the cells were homogenized using a high-pressure homogenizer. The supernatant was collected after centrifugation at 7,104× *g* for 10 min and applied to a nickel column (1.6 × 15 cm^2^) that had been equilibrated with buffer A. After that, the column was washed with buffer, and the samples were eluted with buffer B (30 g NaCl, 13.3 mL Tris-HCl (pH 7.5), 30 g imidazole, and 1 mL Tween-20; HCl was used to adjust the pH to 7.5, and 1 L of distilled water was added). Then, the eluent was further separated through a Superdex G200 column (1.6 × 90 cm²), which has been equilibrated with 1× phosphate buffer saline (PBS) buffer, and the collected samples were analyzed by sodium dodecyl sulfate—polyacrylamide gel electrophoresis (SDS–PAGE).

### 2.6. Protein Quantification

The standard bicinchoninic acid (BCA) was prepared as a 1 mg/mL standard and diluted into 800, 400, 200, 100, 50, and 0 µg/mL BCA standards using ddH_2_O, and 10 µL of each was added to a 96-well plate. The protein samples were diluted 2-fold, 5-fold, and 10-fold, and 10 µL of the stock solution was added to the sample wells of a 96-well plate. Into each well of the standard curve and sample wells, 300 µL of the working solution was added and then the plate was stored at 25 °C for 10 min protected from light. Values at 562 nm were detected using a microplate reader, and a standard curve was plotted. *R*^2^ > 0.99 was considered valid for the standard curve. Protein content was calculated from a valid standard curve and the *R*^2^ formula.

### 2.7. Sugar Quantification

Preparation of standard curve: first, ddH_2_O was used to prepare the glucose mother solution at a concentration of 100 mg/mL. Then, in seven dry and clean glass test tubes, the glucose mother solution and ddH_2_O were used to prepare glucose solutions at concentrations of 0, 10, 20, 40, 60, 80, and 100 µg/mL. All the solutions were quantified to 250 µL in each tube. The protein samples to be tested were diluted to 2-fold, 5-fold, and 10-fold the original confrontation and also quantified to 250 µL in each dry and clean tube. A sufficient quantity of sulfuric acid–anthrone solution was prepared in a ratio of 2 mg anthrone to 1 mL sulfuric acid. To each tube, 1 mL of sulfuric acid–anthrone solution was added and allowed to cool on ice. All tubes were placed in a boiling water bath for 10 min at the same time, and a significant color change could be observed. The tubes were allowed to cool to room temperature on ice. To a 96-well plate, 200 µL of each tube was added, the value was read at 620 nm using a microplate reader, and a standard curve was drawn. *R*^2^ > 0.99 indicated that the curve was reliable, and the sugar content of the experimental group was calculated using the standard curve.

### 2.8. Transmission Electron Microscopy

Pure protein samples were selected and diluted to 100, 50, and 20 µg/mL using sterile 1× PBS. After the samples were mixed, they were slowly dropped on a 200-mesh copper mesh and allowed to stand for 3 min, and then filter paper was used to remove the excess samples. Negative staining solution (2% uranyl acetate) was dropped and allowed to air-dry for 1 min; the excess staining solution was removed using filter paper. The field of view was observed using a Hitachi HT7700 microscope (voltage 80 kV), and the appropriate scale field was selected for photography and preservation.

### 2.9. Dynamic Light Scattering

The protein samples were centrifuged at 16,000× *g* for 10 min, the supernatant was aspirated, and the samples were diluted to a concentration of approximately 200 µg/mL protein using sterile 1× PBS. Into a 2 mL disposable cuvette, 1 mL of the protein samples was pipetted and then the cuvette was placed into the dynamic light scattering instrument detection tank for detection, with each sample tested in triplicate.

### 2.10. SpyCatcher/SpyTag Binding

After KPO1-SC or KPO2-SC and ST-AP205 were calculated by protein quantification, they were mixed overnight at 4 °C, and the binding product was analyzed by SDS–PAGE. Formulas: ST-AP205 (µg/mL): SpyTag molecular weight/ST-AP205 molecular weight × protein concentration; KPO1-SC or KPO2-SC (µg/mL): SpyCatcher molecular weight/KPO1-SC (KPO2-SC) molecular weight × protein concentration.

### 2.11. Statistical Analysis

All analyses were performed using GraphPad Prism 8.0 statistical software (GraphPad Inc., San Diego, CA, USA). Data were analyzed using the *t* test or one-way ANOVA with Dunn’s multiple comparisons test. The results were expressed as mean ± standard deviation (SD). Values of *p* < 0.05 were considered statistically significant.

## 3. Results

### 3.1. Bioinformatics Analysis of Virulence Factors in K. pneumoniae

Common virulence factors in *K. pneumoniae* include capsule, lipopolysaccharide, fimbriae, outer membrane proteins, siderophores, and urease. The whole genome of *K. pneumoniae* ATCC 25955 was downloaded from ATCC to analyze (ATCC® 25955™ [30]) and confirm the presence of a complete capsule biosynthesis gene cluster (*galF*–*gnd*) and a lipopolysaccharide gene cluster (*lpxK*–*coaD*). There was a paucity of fimbriae-related genes, with only *fimH* and *fimK* remaining. *fimH* acts as an adhesin and assembles into fiber-like tip structures in Enterobacteriaceae, whereas *fimK* regulates *fimS* expression. However, these two genes alone are insufficient to generate fimbriae structures. Four siderophores including enterobactin, yersiniabactin, aerobactin, and colibactin were detected in the strain, with only enterobactin being the predominant one. Studies have indicated that enterobactin is essential for bacterial growth and survival. Two major outer membrane proteins, OmpK35 and OmpK36, were also identified, and their knockout results in the defective growth of strains, thus affecting survival. Therefore, the glycoengineering of *K. pneumoniae* ATCC 25955 mainly stems from the absence of lipopolysaccharide (LPS) and CPS.

### 3.2. Glycoengineering of K. pneumoniae Chassis Using CRISPR/Cas9

The CPS can help the bacterium evade the killing effect of host phagocytic cells, increase its survival rate, and ensure its virulence. Genes related to capsular synthesis are clustered in the genome of K. pneumoniae, forming a gene cluster. LPSs, also known as endotoxins, could trigger bacterial septic shock. WaaL is the O-antigen ligase, responsible for transferring the OPS to the lipid A-core. Deletion of the waaL gene not only prevents the production of LPS in K. pneumoniae but also provides more glycans for the production of target glycoproteins.

Therefore, CRISPR/Cas9 tools were used to construct a series of scarless gene deletion mutants derived from the K. pneumoniae wild-type strain 25955 (Figure 1a). First, two single gene mutants, 25955∆waaL and 25955∆CPS, were obtained using the pCas/pTargetF system as described by Jiang et al. All of the gene deletion mutants were verified by sequencing (Figure 1b). To test the deletion efficiency of the CRISPR system for different N20 sites, 4 N20 sites for the waaL gene (named A-D) and 10 N20 sites (named A-J) for the CPS gene cluster were designed and evaluated in this study. Verification of correct clones with target gene deletion was achieved by performing PCR. The results indicated that the selection of N20 sites had a significant effect on gene deletion and varied from 0% to 66% (Figure 1c). However, the concatenation of two N20 sites with the highest deletion efficiency (I, 64%; G, 58%) did not improve the efficiency of deletion further (bars G-I and I-G in Figure 1c). Based on this, the CPS gene cluster from strain 25955∆waaL was further deleted to obtain the double gene mutant 25955∆waaL∆CPS (Figure 1d) for the subsequent experiments.

### 3.3. Characteristics of Glycoengineered 25955 Strains

After a series of scarless deletion mutants were obtained (25955∆waaL, 25955∆CPS, and 25955∆waaL∆CPS), the growth curve, viable counts, settling velocity morphology, and carbohydrate catabolism of these bacteria were determined to test whether the gene deletions have adverse effects on bacterial growth. Although there was no significant difference in the number of viable bacteria, bacterial growth rates decreased slightly after double deletions (Figure 2a). Notably, bacterial flocculation occurred when mutant strain 25955∆waaL∆CPS was incubated at room temperature in LB medium for 6 h (Figure 2b). This phenomenon might be caused by the general lack of hydrophilic surface glycans after the deletion of the two genes. In addition, there was no significant difference in the bacterial morphology detected by transmission electron microscopy and carbohydrate catabolism (Figure 2c,d).

### 3.4. The Glycoengineered 25955∆waaL∆CPS Chassis Produced K. pneumoniae O1 Glycoprotein

To demonstrate the practical value of the engineered bacteria constructed in this study, 25955∆waaL∆CPS was used as the chassis for the production of K. pneumoniae O1 glycoproteins (KPO1-SC). A plasmid encoding the glycosyltransferase PglL and the carrier protein SC4573 (pET28a-SpyCatcher4573) [27] was electroporated into this chassis (Figure 3a). Further, after induction by IPTG (final concentration 1 mM), the expression of targeted KPO1-SC glycoproteins by the 25955∆waaL∆CPS chassis was detected by SDS–PAGE and Western blot. The result was indicated by a classic ladder pattern on gel electrophoresis, which could be recognized by antibodies against the O1 OPS (Figure 3b), indicating that the target glycoproteins could be produced by our glycoengineering strain. The average chain length of glycan chains was revealed to be approximately 55 kDa for KPO1-SC according to the western blot, and the polysaccharide–protein ratio was 1:2.86 (estimated) based on sugar quantification and protein quantification results. 

### 3.5. The Influence of ECA Synthesis on the KPO1-SC Glycoprotein Yields

A good K. pneumoniae chassis for PGCT requires excluding competition from the endogenous glycan synthesis pathway to facilitate the coupling of exogenous antigenic glycans to carrier proteins to produce targeted glycoproteins. Therefore, CRISPR/Cas9 tools were used to delete the enterobacterial common antigen (ECA) gene cluster from the 25955∆waaL∆CPS chassis (Figure 4a). When incubated at room temperature for 6 h, bacterial flocculation of the triple deletion mutants (25955∆waaL∆CPS∆ECA) could be observed more distinctly than that of its parent strain (25955∆waaL∆CPS) (Figure 4a). However, it demonstrated no significant difference in the growth curve and viable counts (Figure 4b). The deletion efficiency of the CRISPR system with different N20 sites was also determined as described earlier. Twelve N20 sites for the ECA gene cluster were targeted. The results indicated that the gene deletion efficiency varied from 0% to 41% (Figure 4c). Then, the aforementioned plasmid (pET28a-SpyCatcher4573) was electroporated into the 25955∆waaL∆CPS∆ECA strain to verify the expressions of target glycoproteins. The results indicated that with further deletion of ECA, glycoprotein expression was virtually abandoned (Figure 4d). Thus, the double mutant 25955∆waaL∆CPS was selected as the chassis for the production of glycoproteins.

### 3.6. O-Antigen Serotype Shift of the Glycoengineered Chassis

Compared with that of serotype O2 polysaccharide, the polysaccharide of K. pneumoniae strain O1 has a longer chain and more complex structure. The O1 antigenic polysaccharide synthesis gene cluster (wzm, wzt, wbbM, glf, wbbN, wbbO, wbbY, and wbbZ) has two additional genes (wbbY and wbbZ) compared with that of O2 strain. To expand the application range of our chassis strain, we aimed to further express O2 serotype glycoproteins (KPO2-SC) using the former strain 25955∆waaL∆CPS (O1 serotype) (Figure 5a). Then, CRISPR/Cas9 tools were used to delete the wbbY-Z gene cluster (Figure 5b). The triple deletion mutants demonstrated no significant difference in the growth curve and viable counts when compared with the parent strain (Figure 5c). Five N20 sites for the wbbY-Z gene cluster were targeted and tested, and the deletion efficacy varied from 0% to 41% (Figure 5d). Then, plasmid pET28a-SpyCatcher4573 was electroporated into the 25955∆waaL∆CPS∆wbbY-Z strain to verify the expression of KPO2-SC. The expression of the targeted KPO2-SC glycoproteins in the 25955∆waaL∆CPS∆wbbY-Z chassis was detected by SDS–PAGE and Western blot (Figure 5e). In addition, the samples containing glycoproteins KPO1-SC and KPO2-SC were analyzed simultaneously to rule out the cross-reaction between anti-O1 and anti-O2 sera (Figure 5f).

### 3.7. Preparation of Bioconjugate Nanovaccines Using the Glycoproteins Produced by Glycoengineered Chassis

The 25955∆waaL∆CPS strain and 25955∆waaL∆CPS∆wbbY-Z strain were used as chassis for the production of K. pneumoniae O1 and O2 glycoproteins (KPO1-SC and KPO2-SC) respectively. The purified KPO1-SC and KPO2-SC were combined with SpyTag-Ap205 (ST-VLP) nanoparticles, which have been described by our previous work [27], to prepare bioconjugate nanovaccines (Figure 6a). The resulting nanovaccines KPO1-VLPs and KPO2-VLPs (Figure 6b) were obtained after overnight binding in 1× PBS at 4 °C. The particle size of the nanovaccines was homogeneous as demonstrated by dynamic light scattering analysis (Figure 6c) and transmission electron microscopy imaging (Figure 6d). However, the purification of the final product remains a problem to be solved because there were still some unbound protein nanoparticles after size exclusion chromatography separation.

## 4. Discussion

*K. pneumoniae* polysaccharide biosynthesis gene clusters of serotypes O1 and O2 were only differentiated by two genes (*wbbY* and *wbbZ*). This encouraged us to use a single bacterium (strain 25955) to construct two glycoengineered chassis to express different O-serotype glycoproteins. This approach reduces the costs of host engineering and paves the way for the production of bivalent *K. pneumoniae* vaccines in the future. Unfortunately, the purity of the final product did not meet our expectations in this study, as multiple impurities were observed, which limits its industrial application in its current state. Proteomic analysis methods can be used to investigate the protein sequences of these impurities. Improved purification methods, such as changing the purification tag, using more precise column packing materials, and multiple purifications could be utilized to eliminate impurities.

The production of *K. pneumoniae* glycoproteins involves a wide range of genes in polysaccharide biosynthesis, including nucleotide sugar precursor synthesis genes, glycosyltransferase genes, and polysaccharide processing genes (such as those involved in side-chain modification). Host cells often express many other glycopolymers in addition to the target exogenous polysaccharides. These non-targeted glycopolymer synthesis pathways may interfere with the synthesis and assembly of the target polysaccharide antigen by competing for lipid carriers and activating monosaccharide substrates. Therefore, the genetic background and the basic metabolic pathways of *K. pneumoniae* 25955 need to be further clarified using bioinformatics and systems biology analyses. Removing the nonessential polysaccharide biosynthesis gene cluster will facilitate the correct and efficient expression of the target polysaccharide antigen. In this study, we attempted to eliminate unnecessary glycan interference by removing the endogenous, nonessential ECA biosynthetic gene cluster. However, deleting the ECA biosynthesis gene cluster did not lead to the expected increase in glycoprotein production. One possible explanation is that the deletion of ECA clusters eliminates an essential gene for target polysaccharide biosynthesis. However, further experiments, such as gene complement experiments, are needed to verify this hypothesis.

*E. coli*, a model strain for studying microbial genetics, physiology, and metabolism, has become an important chassis because of its advantages in genetic manipulation tools and clear genetic background. Various modified *E. coli* strains have been used as hosts for producing various compounds. Recently, a glycoengineered *E. coli* chassis for PGCT platforms has been used to produce *K. pneumoniae* capsular polysaccharide conjugate vaccines. It can be inferred that O-serotype polysaccharide conjugate vaccines could also be prepared using *E. coli* as a host strain by introducing a gene cluster for the heterologous biosynthesis of *K. pneumoniae* OPSs. In comparison, our *K. pneumoniae* glycoengineered chassis required only one plasmid encoded by the glycosyltransferase and carrier protein to complete glycoprotein production, reducing the use of antibiotics and improving plasmid stability [31,32]. Notably, the problem caused by the use of multiple plasmids can also be solved by integrating key components (genes) into the genome of the host as presented in recent studies. Therefore, we can further abolish the use of any plasmid by integrating the genes encoding PglL and SC4573 proteins into the genome of our *K. pneumoniae* glycoengineered chassis in the next step.

It should also be noted that *K. pneumoniae*, as a chassis for bioengineering, has the obvious advantage of being able to utilize numerous carbon sources, particularly cost-efficient crude glycerol, leading to better economic value. Thus, the cost-efficient culture medium and the optimal fermentation conditions should also be evaluated to increase glycoprotein yields. Furthermore, more efficient gene editing techniques can be used to further simplify the genome of *K. pneumoniae* chassis to reduce the metabolic burden, increase the growth rate, and promote the production of glycoproteins.

## 5. Conclusions

In this study, two glycoengineered chassis with different O-serotypes were constructed following the deletion of the CPS gene cluster and *waaL* gene in *K. pneumoniae* strain 25955. KPO1-SC and KPO2-SC glycoproteins were successfully expressed and purified in the glycoengineered chassis strains by using the recombinant SpyCatcher protein fusion with 4573 sequons (recognized by *O*-glycosyltransferase PglL) as the carrier protein in PGCT methods. In addition, KPO1-VLPs and KPO2-VLPs were successfully constructed in vitro by covalently linking SC protein with SpyTag decorated on the surface of AP205 nanoparticles. In summary, we have constructed two glycoengineered *K. pneumoniae* strains with different O-antigen serotypes as a complement to the current glycoengineering chassis strain library for PGCT methods. This study provides an effective solution for preparing *K. pneumoniae* nanovaccines, and the strategy is expected to be extensively used in future synthetic biological research.

## Figures and Tables

**Figure 1 microorganisms-11-01321-f001:**
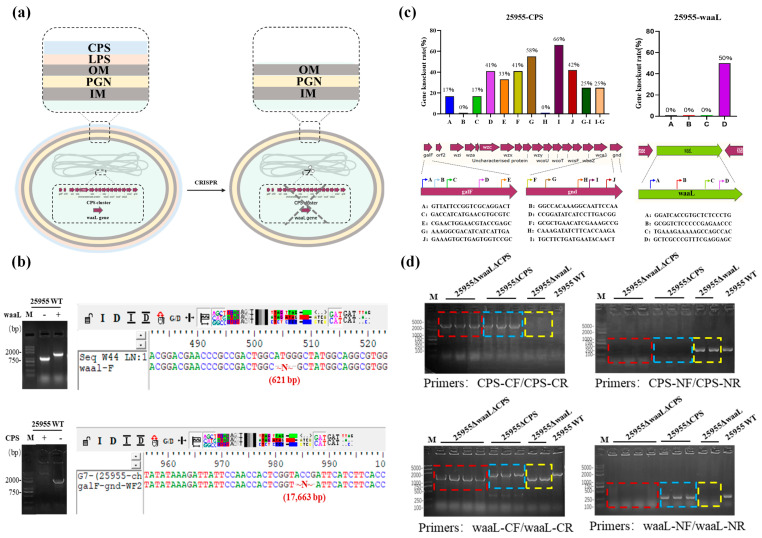
Deletion of two major virulence factors in Klebsiella pneumoniae ATCC 25955. (**a**) Schematic representation of LPS and CPS gene cluster scarless deletion mutants derived from K. pneumoniae wild-type strain 25955. (**b**) PCR and sequencing validation of the deletions of the waaL gene and CPS gene cluster. (**c**) Verification of deletion efficiency using different N20 sites. A total of 12 clones from each N20 site were tested, and the arrow directions indicate the relative position of N20 and the corresponding PAM. (**d**) PCR validation of the constructed 25955 chassis strain (deletion of both the waaL gene and the CPS gene cluster). CPS deletion strains were detected using CPS-CF/CR primers, and CPS non-deletion strains were detected using CPS-NF/NR primers. waaL deletion strains were detected using waaL-CF/CR primers (non-deletion strains exhibited higher bands than deletion strains), and waaL non-deletion strains were detected using CPS-NF/NR primers. LPS, lipopolysaccharide; CPS, capsule polysaccharide; PAM, protospacer adjacent motif. (red dotted: 25955∆waaL∆CPS chassis strain; blue dotted: 25955∆CPS chassis strain; yellow dotted: 25955∆waaL chassis strain).

**Figure 2 microorganisms-11-01321-f002:**
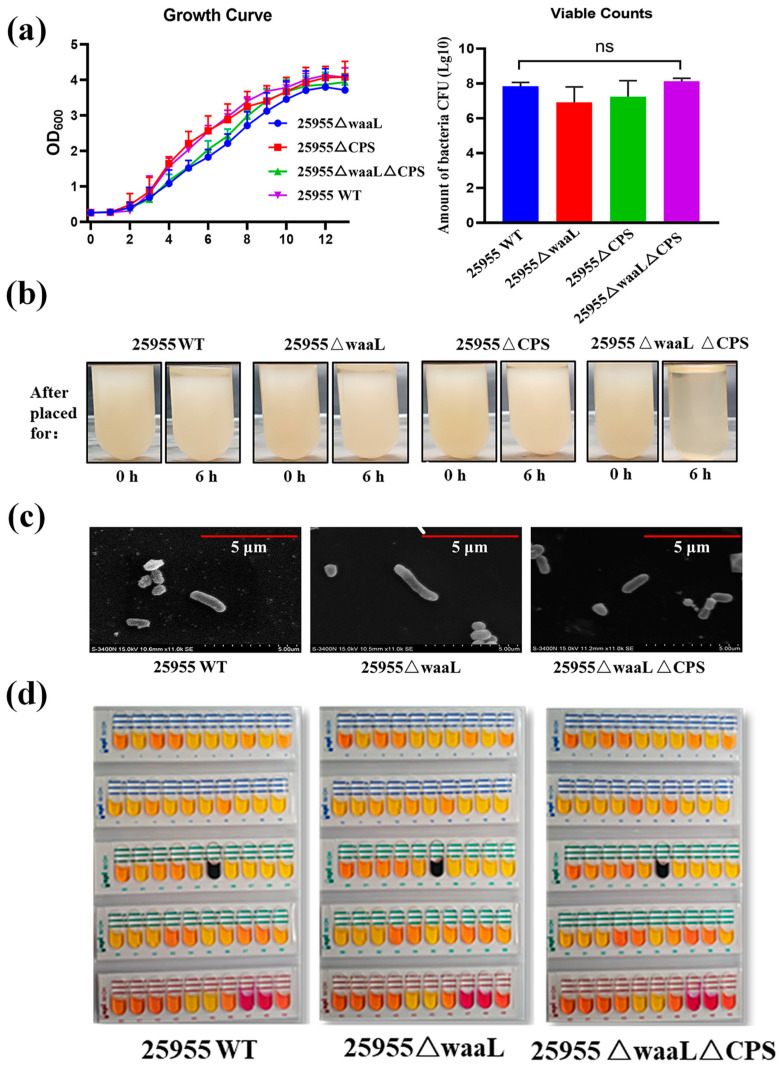
Characterization for 25955 chassis strain. (**a**) Growth curves and viable counts of 25955 wild-type strain and each mutant strain (25955∆waaL, 25955∆CPS, and 25955∆waaL∆CPS). Each strain was inoculated with a seed solution at OD_600_ = 2.0 and with a ratio of 1:100; the viable bacteria were counted by gradient dilution coating on LB plates at OD_600_ = 2.0. (**b**) Settling velocity of 25955 wild-type strain and each mutant strain (25955∆waaL, 25955∆CPS, and 25955∆waaL∆CPS). (**c**) TEM images of 25955 wild-type strain and each mutant strain (25955∆waaL and 25955∆waaL∆CPS). Scale bar, 5 μm. (**d**) Validation of the carbohydrate metabolism of 25955 wild-type strain and each mutant strain (25955∆waaL and 25955∆waaL∆CPS). LB, Luria–Bertani; TEM, transmission electron microscopy.

**Figure 3 microorganisms-11-01321-f003:**
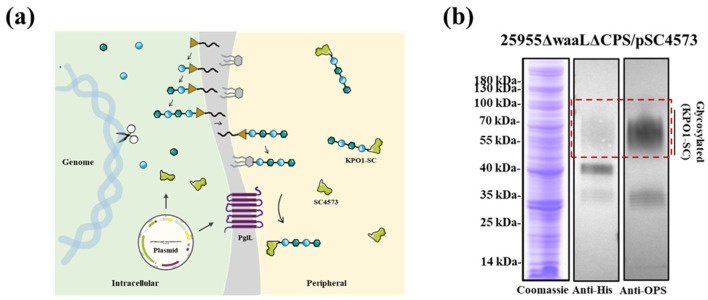
Detection of KPO1-SC glycoprotein expression in chassis strain 25955∆waaL∆CPS. (**a**) Schematic representation of KPO1-SC glycoprotein expression. (**b**) The glycoprotein located at the higher molecular weight region is marked by a red dotted box. KPO1-SC glycoproteins produced by strain 25955∆waaL∆CPS were analyzed by SDS–PAGE and WB with anti-His and anti-OPS (O1) antibodies. WB, Western blot (The color of the His antibody versus the serum antibody WB exposed is determined by the exposure time. Because there are some miscellaneous proteins, His antibody interferes with the determination of glycoprotein when the exposure time is long, whereas serum antibody does not detect carrier proteins, so we selected images with long exposure time and clear glycoprotein position).

**Figure 4 microorganisms-11-01321-f004:**
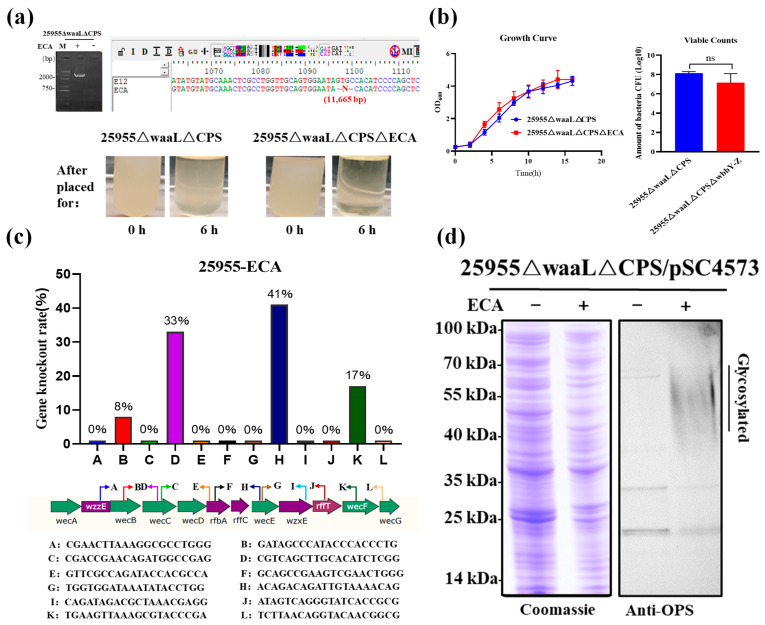
Deletion of the ECA cluster in chassis strain 25955∆waaL∆CPS. (**a**) Deletion of the ECA cluster was confirmed by PCR, sequencing validation, and settling velocity. (**b**) Growth curves and viable counts of 25955∆waaL∆CPS strain and 25955∆waaL∆CPS∆ECA strain. Each strain was inoculated with a seed solution at OD_600_ = 2.0 with a ratio of 1:100; the viable bacteria were counted by gradient dilution coating on LB plates at OD_600_ = 2.0. (**c**) Verification of deletion efficiency of ECA cluster with different N20 sites. A total of 12 clones were tested from each N20 site, and the arrow directions indicate the relative positions of N20 and corresponding PAM. (**d**) Comparison of KPO1-SC glycoprotein expression before and after deletion of the ECA cluster in strain 25955∆waaL∆CPS. KPO1-SC glycoproteins produced by strain 25955∆waaL∆CPS and 25955∆waaL∆CPS∆ECA were analyzed by SDS–PAGE and WB with anti-OPS (O1) antibodies. ECA, enterobacterial common antigen; LB, Luria–Bertani; PAM, protospacer adjacent motif; WB, Western blot.

**Figure 5 microorganisms-11-01321-f005:**
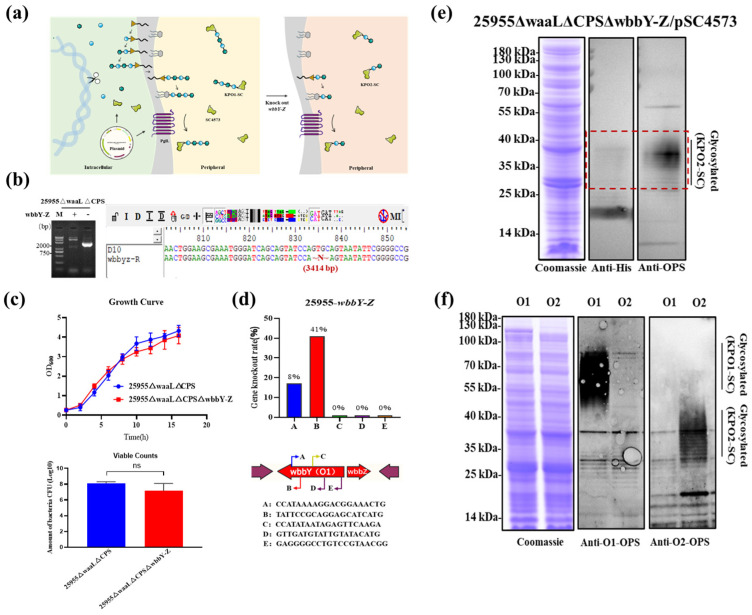
Detection of KPO2-SC glycoprotein expression in chassis strain 25955∆waaL∆CPS∆wbbY-Z. (**a**) Schematic representation of KPO2-SC glycoprotein expression. (**b**) Deletion of the wbbY-Z cluster was confirmed by PCR and sequencing validation. (**c**) Growth curves and viable counts of 25955∆waaL∆CPS strain and 25955∆waaL∆CPS∆wbbY-Z strain. Each strain was inoculated with a seed solution at OD_600_ = 2.0 and with a ratio of 1:100; the viable bacteria were counted by gradient dilution coating on LB plates at OD_600_ = 2.0. (**d**) Verification of wbbY-Z cluster deletion efficiency of different N20 sites. A total of 12 clones were tested from each N20 site, and the arrow direction indicates the relative position of N20 and the PAM. (**e**) KPO2-SC glycoproteins produced by strain 25955∆waaL∆CPS∆wbbY-Z were analyzed by SDS–PAGE and WB with anti-His and anti-OPS (O2) antibodies. The glycoprotein located at the higher molecular weight region is marked by a red dotted box. (**f**) Evaluation of the cross-reaction of immune serum against the OPS (O1) and OPS (O2) by SDS–PAGE and WB with anti-OPS (O1) and anti-OPS (O2) antibodies. LB, Luria–Bertani; PAM, protospacer adjacent motif; WB, Western blot.

**Figure 6 microorganisms-11-01321-f006:**
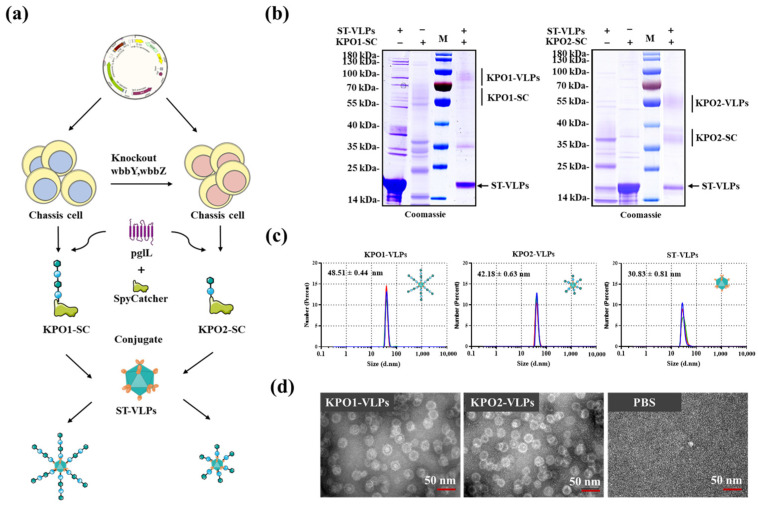
Preparation of conjugate nanovaccines. (**a**) Schematic diagram of SpyCatcher/SpyTag nanoparticle assembly. (**b**) SDS–PAGE analysis of the binding of KPO1-SC/KPO2-SC and ST-VLPs nanoparticles after overnight incubation at 4 °C. (**c**) DLS assays were performed to analyze KPO1-VLP, KPO2-VLP, and ST-VLP nanoparticles. (**d**) TEM images of KPO1-VLP and KPO2-VLP nanoparticles. Scale bar, 50 nm. DLS, dynamic light scattering; TEM, transmission electron microscopy.

**Table 1 microorganisms-11-01321-t001:** *Klebsiella pneumoniae* strains and plasmids used in this study.

Strains and Plasmids	Genotype Descriptions	Reference
Strains		
*K. pneumoniae ATCC 25955*	25955 wild-type strain	[26]
25955∆*waaL*	*waaL* gene was knocked out in the 25955 strain	This work
25955∆CPS	CPS gene cluster was knocked out in the 25955 strain	This work
25955∆*waaL*∆CPS	CPS gene cluster was knocked out in 25955∆*waaL*	This work
25955∆*waaL*∆CPS∆ECA	ECA gene cluster was knocked out in 25955∆*waaL*∆CPS	This work
25955∆*waaL*∆CPS∆wbbY-Z	wbbY-Z gene cluster was knocked out in 25955∆*waaL*∆CPS	This work
Plasmids		
pET28a-SpyCatCher4573	Cloning SpyCatCher4573 gene in pET28a	[27]
pCas	The CRISPR system plasmid	[28]
pTargetF	The CRISPR system plasmid	[28]

## Data Availability

No new data was created.

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
