# Peer review of "Genetic Engineering of Klebsiella pneumoniae ATCC 25955 for Bioconjugate Vaccine Applications"

_microorganisms, 2023, doi:10.3390/microorganisms11051321_

Round 1

Reviewer 1 Report

In the paper “Genetic engineering of Klebsiella pneumoniae ATCC 25955 for 2 bioconjugate vaccine application”  Liu et al. provide new indications for developing bioconjugate nanovaccines using K. pneumoniae as bacterial chassis.

For this aim, the authors take into consideration two serotypes of Klebsiella representing the 60% of clinical isolates resistant to carbapenems.

We found the study enough interesting and complete. We notice that the conclusions overestimate the real value of the results, neglecting the necessary further steps to improve the achievements. Further developments are expected to increase the quality of the final product by investigating the metabolic pathways of K. pneumoniae.

We reported some observations below:

Figures 1d and 3b are cryptic, please integrate more information in the legend.

Line 292: please cite your work.

Line 346: please cite works.

Reviewer 2 Report

Line No

Comment

General

Is there any evidence that antibodies against the LPS O-chain are protective against disease?  Please include a reference for this. Previous vaccines have targeted the CPS, not the LPS O-chain.

General

The focus of this paper is on the genetic changes to produce a specific product, however there is insufficicient information to prove that they have really made what they claim to have made: only a Western Blot to show the presence of the glycan and a DLS method [not described in the Materials and Methods] for homogeneity of molecular size. The molecular size is not calibrated to provide an estimate of molecular weight.

General

The Materials and Methods Section is incomplete, with little or no information describing the methods used: see below.

General

Normal expectations for a glycoconjugate vaccine would include the average chain lengths of the glycan chains, polysaccharide-protein ratio, number of glycan chains per carrier protein etc. Fig. 3 implies a single glycan chain per carrier protein with a single repeat unit. But no experimental evidence to support this. It is inconsistent with the claim of a ladder structure in Western Blot experiments. Some text suggests that there is a significant amount of unconjugated protein present, but no methodology to quantify this is reported, and it would not be detected by Western blotting targetting the glycan chain.

General

No evidence of immunogenicity, which is critical for further development of a glycoconjugate antigen into a vaccine, is presented.

18, 83

What is a "nanovaccine"? The paper would benefit from a clear definition of this term.

47

Jain and Ravinder papers [Refs. 10 and 12] recognised by first author, but not Ref. 11, which is linked to a different author. It seems curious way to do this, and the authors should be consistent in their usage.

36-51 and 63-67

All the papers cited report results with capsular polysaccharide immunogens [free or conjugated], but the authors here describe LPS O-chain conjugates. Is there any evidence antibodies against LPS O-chain are protective?

Materials and Methods

There is lack of description in the methodology, apart from Western blotting, seems to imply that no significant characterisation of the glycoconjugate product was carried out. No information about the DLS or electron microscopy work. There is no information about how carbohydrate catabolism was quantified. Analytical methods should not be described in Figure Legends. There is no information on protein or glycan quantification. The methods used to monitor elution from columns used in protein purification [and identification of required fractions] are not described. The authors should expand this section so that every technique and analytical procedure is adequately described.

Materials and Methods

The authors provide no information about the scale at which the cells were grown [which affects the size of columns used in purification], or the yield of protein and glycan that they achieved. If this material is to become a vaccine manufacturing platform, this information is critical.

205-206

No information in Materials and Methods as to the equipment and methodology used for these experiments. This should be included.

Legend to Fig. 2, Fig. 4, Fig. 5

This information should be in Materials and Methods section.

219-220

What is SC4573 - sequence, any info on previous use as a carrier protein etc.? If the information is in the Supplementary informatrion, then this should be explicit in the text.

Figure 3

The presence of strong bands from anti-His antibodies quite distinct from the anti-LPS bands seems to suggest that a relatively small proportion of the protein produced was glycosylated. Quantification of these two types of material is highly relevant to the reader's interpretation of this work.

292-293

"…. which have been described in our previous work…". The papers should be cited to allow readers to access the information.

Fig. 6

This shows results from dynamic light scattering experiments, but there is no information about the equipment, calibration and methodology in the Materials and Methods Section. The diagrams for Fig. 6(c) imply that there are six glycan chains per carrier protein. Is there any evidence for this, or is it just something to give a pretty picture? Was any molecular sizing carried out, by size exclusion chromatography [SEC], SEC-MALLS or mass spectrometry? FIgure 6(b) shows the separation of the the VLPs on SDS-PAGE. In my experience, SDS treatment leads to disaggregation of non-covalently associated protein aggregates. The claimed stability of the VLPs to SDS treatment deserves comment.

297-298

".. Still some unbound protein nanoparticles after size-exclusion chromatography…." Not bound to what? The authors provide to information. Or do they means not carrying glycan chains? The authors should clarify exactly what they mean.

330

" … did not lead to the expected increase in glycoprotein production…." There is no method reported in the Materials and Method Section to quantify the glycoprotein production. This should be included. Since this was measured, quantitative information in the manuscript would be of interest to the reader.

Some phrasing is difficut to understand

Reviewer 3 Report

Dear Authors,

congratulations for the article; only minor revision are requested:

134 The induced cells were collected by centrifugation at 8000 rpm for 10 min and then 134

137 The supernatant was collected after centrifugation at 8000 rpm for 10 min, and applied on 137

In 134 and 137 you show the number of revolutions but not the type or diameter of the rotor, please indicate the diameter of the rotor or, simply, the force of gravity (G-force). Overall, the work is very interesting and opens up new possible further research above all to define the nature of the impurities and improve the final product (bivalent vaccine against K. pneumoniae).Congratulation for the great job

Round 2

Reviewer 2 Report

Q1 The authors comment on the published work of Myeongjin Choi as demonstrating that antibodies against the OPS are protective, but choose to insert citations which don't address the specific question. Why not just cite the Myeongjin Choi papers?
Q4 Again, the authors respond with information to the reviewer, but do not include this in the manuscript. Why should this information remain hidden from other readers? Please include the data in the publlished manuscript.
Q4 The repeat unit of the K. pneumoniae O1 LPS is a di-galactose repeat, with a molecular mass of 324 Da. I wold be amazed if the gel was able to resolve glycoproteins differing only in the number of repeat units [ie. by 324 Da]. A glycan chain of 55kDa would have well in excess of 100 repeat units. 
Q5 The authors talk around the question asked, but never answer it. So I assume that they have no evidence for the immunogenicity of their construct. A direct response to the question would have been appreciated - rather than wasting my time.
Q10 Thank you for the answer to the question. It would be even better if this information were to be included in the manuscript - either in the main text or by specifying incubation times in the Experimental Section.